# Prospective cohort study of exposure to tobacco imagery in popular films and smoking uptake among children in southern India

**Muralidhar M. Kulkarni**[1]**, Asha Kamath** [2]*****, Veena G. Kamath**[1]**, Sarah Lewis**[3]**, Ilze Bogdanovica**[3]**, Manpreet Bains**[3]**, Jo Cranwell**[4]**, Andrew Fogarty**[3]**, Monika Arora**[5,6]**, Gaurang P. Nazar**[5,6]**, Kirthinath Ballal**[1]**, Ashwath K. Naik**[2]**, Rohith Bhagawath**[1]**, John Britton**[3]

**1** Community Medicine, Kasturba Medical College, Manipal Academy of Higher Education, Manipal, Udupi, Karnataka, India, **2** Department of Data Science, Prasanna School of Public Health, Manipal Academy of Higher Education, Udupi, Karnataka, India, **3** UK Centre for Tobacco and Alcohol Studies, University of Nottingham, Nottingham, United Kingdom, **4** Department for Health, Tobacco Control Research Group, University of Bath, Bath, United Kingdom, **5** HRIDAY, Delhi, India, **6** Health Promotion Division, Public Health Foundation of India, Delhi, India

* asha.kamath@manipal.edu

**Data Availability Statement:** Data cannot be shared publicly because of constraints of the ethics committee. Data are available from the MAHE

## Abstract

### Background

Exposure to tobacco imagery in films causes young people to start smoking. Popular Indian films contain high levels of tobacco imagery, but those that do are required by law to display onscreen health warnings when smoking imagery occurs and to include other health promotion messaging before and during the film. We report a prospective cohort study of incident smoking in relation to exposure to film tobacco imagery and anti-tobacco messaging in a cohort of children in southern India.

### Methods

We carried out a one-year longitudinal follow up questionnaire survey in 2018 of a cohort of 39,282 students in grades 6, 7 and 8 (aged between 10 and 15 years) in schools in the Udupi district of Karnataka State in India who participated in a 2017 cross-sectional study of exposure to smoking in films and ever smoking status.

### Results

We obtained usable linked data in 2018 from 33,725 of the 39,282 (86%) participants with data from 2017. Incident smoking was reported by 382 (1.1%) participants. After adjusting for age, sex and common confounders significantly associated with incident smoking there was no significant independent effect of exposure to film smoking, either as a binary (Odds Ratio 1.6, 95% Confidence Interval (CI) 0.5 to 4.9) or as a graded variable, on smoking uptake. An exploratory analysis indicated that the presence of on-screen health warnings that complied fully with Indian law was associated with a significantly lower odds of smoking uptake (Odds Ratio 0.8 (0.6 to 1.0, p = 0.031) relative to the same exposure sustained in absence of compliant warnings.

Institutional Data Access / Ethics Committee (contact via mrcuk.antitobacco@manipal.edu) for researchers who meet the criteria for access to confidential data.

**Funding:** The authors are fulltime employees of their respective institutions and draw incentives to compensate for the time through the grant. RB is the Social Scientist in the project and gets his salary from the project. The authors acknowledge the support of the Medical Research Council through grant MR/P008933/1 to JB. The funders had no role in study design, data collection and analysis, decision to publish, or preparation of the manuscript.

**Competing interests:** The authors have declared that no competing interests exist.

## Conclusion

Exposure to tobacco imagery in Indian films was not associated with a significantly increased risk of incident smoking in South Indian children. While it is possible that this finding is a false negative, it is also possible that the effect of film exposure has been attenuated by the presence of on-screen health warnings or other Indian tobacco-free film rules. Our findings therefore support the wider implementation of similar tobacco-free film measures in other countries.

## Introduction

Preventing smoking uptake is crucial to global public health, and particularly so in the low and middle income countries where the majority of the world's current and likely future smokers live [1,2]. One important and largely preventable cause of smoking uptake is exposure to smoking imagery in feature films [3–6], which has been shown in meta-analysis of prospective studies to increase the risk of smoking uptake by over 40% [7]. Exposure of children to smoking imagery in films can be reduced by measures such as prohibiting paid product placement or inclusion of brand imagery in new films, and requiring films containing smoking to be given adult age ratings. However, these measures do not prevent filmmakers from including tobacco imagery without payment, or protect children who access or are shown adult-rated films, or the vast library of films released before such measures came into force. For this reason, Article 13 of the World Health Organisation Framework Convention on Tobacco Control recommends the display of anti-tobacco messaging before the beginning of any entertainment media product that includes tobacco imagery [8]. However, while showing anti-tobacco messaging before films containing smoking has been shown to generate negative perceptions of smoking among exposed young people [9,10], there is no evidence on the effectiveness of this and other measures to reduce the impact of exposure on smoking uptake from prospective studies.

In 2012, the Indian government introduced the tobacco-free film and TV Rules (mentioned as 'Rules' hereafter) requiring that screenings of films containing tobacco imagery include an audio-visual disclaimer at the start and during the film, anti-smoking 'health spots' before and during the film, and on-screen health warnings during scenes containing tobacco [11]. We have recently reported cross-sectional evidence that, despite the significant non-compliance with these legal requirements and contrary to findings from elsewhere in the world and from India before 2012, children exposed to tobacco imagery in popular films in Karnataka State in southern India in 2017 were not at increased risk of smoking, suggesting that these Rules may be effective [12]. We now report a one-year prospective cohort study of incident smoking in relation to exposure to film tobacco imagery and anti-tobacco messaging in this cohort of children.

## Methods

We carried out a one-year longitudinal follow up questionnaire survey of a cohort of 39,282 students in grades 6, 7 and 8 (aged between 10 and 15 years) in the more than 700 government, 250 government-aided and 200 privately-funded schools in Udupi district of Karnataka State in India who participated in a previously reported cross-sectional study of the association between exposure to smoking in films and ever-smoking in 2017 [12]. In 2018 we re-surveyed

these participants by using the same procedures as previously described [12] to recruit schools, distribute information sheets to students and parents, obtain consent and distribute follow-up questionnaires to all students now in grades 7 to 9. As school attendance rates are high, we studied only those children present on the arranged study day; if for any reason (for example, heavy monsoon rains) fewer than 80% of students were in attendance, the survey was rescheduled. Ethics approvals were granted by the Manipal Academy of Higher Education [MAHE EC/012/2017], Nottingham University [OVS200317] ethics committees, Centre for Chronic Disease Control [CCDC_IEC_11_2018] and the Indian Health Ministry's Screening Committee [HMSC 2017–0460].

## Questionnaire design and study variables

The questionnaire was essentially as described in our previous study [12], eliciting information on current and past use of cigarettes, beedis and a range of other smoked tobacco including cigars, cheroots, chillum, chutta and rolled cigarettes [13], with frequency of use (never; ever but not now; less than once a week; once a week; daily) using questions adapted from the Global Youth Tobacco Survey [14], the UK Smoking, Drinking and Drug Use (SDD) survey [15] and HRIDAY's Mobilizing Youth for Tobacco-Related Initiatives in India (MYTRI) project [16]. (Questionnaire of Year one and Year two are provided as supplementary files in S1 File) Questions on exposure to tobacco imagery in films asked students at baseline in 2017 whether they had seen any of 27 of the most popular films in Karnataka in 2015 and 2016 previously demonstrated to include smoking imagery [11]. The identification and content coding of these films has been reported in detail elsewhere [11] but in brief, we used national and local box office and distributor takings to identify 47 of the most popular films in Karnataka in the given years, and used 5-minute interval coding to provide semi-quantitative estimates of tobacco content in them. We then selected the 27 films found to contain smoking imagery for inclusion in the questionnaires, and asked all participants to report whether they had seen each of these films. Total exposure to tobacco imagery was estimated by summing the number of 5-minute coded intervals containing tobacco imagery contained in each of the films and assuming one complete viewing per film reported as seen. Film compliance with legal requirements under the Indian Rules [17] regarding the inclusion of audio-visual disclaimer, health spots before and during the film and on-screen health warnings during scenes including smoking was also coded [11]. Questions were included on the smoking policies adopted by the respondent's school and in the family home, and on family smoking, peer smoking, self-esteem and rebelliousness [18–20]. We measured socio-economic status through a question on ownership of household goods, grouping participants into quintiles of family wealth [21]. Other variables included age, gender, religion, academic grades in the past year, expectation of academic achievement, and parental education and occupation. A full description of all variables is provided in S1 Annexure (Supplementary file). The questionnaire was piloted in a school in the neighbouring district and refined before use.

## Sample size calculation

With approximately 45,000 potentially eligible participants of the relevant age in the district, and assuming 70% participation in the first survey and 70% retention in the second survey, when we designed our study we anticipated obtaining cross-sectional data in the first survey from approximately 31,000 students, and paired data after the second survey in 22,000. Assuming smoking uptake in approximately 2% of participants, and with a design effect of 1.5 to allow for clustering, these numbers were expected to provide approximately 90% power to detect an odds ratio of 1.5 for the effect of a risk factor for smoking uptake with exposure in 25% of the population.

## Data analysis

Data were extracted from completed questionnaires into Microsoft Excel using Optical Mark Reader scanning and transferred into STATA 9.2 software for analysis. Questionnaires from the two surveys were linked using each student's unique enrolment number, and by using other questionnaire responses (for example, age, school, grade) to confirm matches in the event of minor coding discrepancies. Ever smoking was defined as any reported smoking of any tobacco product, currently or in the past, and incident smoking the reporting of ever smoking in 2018 (year 2) among participants who reported that they had never smoked in 2017 (year 1). Associations between incident smoking and ordered or categorical variables were evaluated using logistic regression to estimate the effects of potential explanatory variables measured in year one on the risk of smoking uptake in year two. We estimated the effects of demographic and other potential confounders first as single variables, then in a model containing all variables of interest, and then in a model retaining age, sex and those confounders that were independently significant (p<0.05) in multivariate analysis. We then tested the effect of exposure to films containing smoking imagery before and after correcting for these independently significant variables, first treating smoking imagery exposure as a binary (yes or no) exposure and then, given that almost all children were exposed to some degree, as a graded exposure with four categories (none, and ordered tertiles) of exposure to smoking intervals in films. In a *post hoc* analysis we then explored the effects of exposure to film smoking intervals in relation to the degree of film compliance with smoke-free rules, adjusting for the same potential confounders. Films were categorised as compliant or non-compliant in relation to AV disclaimers and health spots at the start and middle of the films. On-screen health warnings were analysed in two binary categorisations: health warnings present but not necessarily fully compliant with rules on colour and legibility or not; and health warnings present and fully compliant with COTPA rules, or not.

## Results

As previously reported, in year one, 39,282 of 46,706 (84%) students enrolled in grades 6 to 8 in 914 schools provided questionnaire responses sufficiently complete for analysis [12]. In year two, there were a total of 47,130 students enrolled in grades 7 to 9 in 904 schools (14 schools had closed and 4 new schools opened between studies), of whom 41,664 (88%) provided responses sufficiently complete for analysis. We were able to link year two data with year one responses for 34,414 students, and after excluding responses from 689 individuals with incompatible ever-smoking responses (positive in year one and negative in year two) were left with 33,725 paired responses suitable for analysis. A more complete breakdown of the genesis of these figures is presented (Fig 1).

Incident smoking between years one and two was reported by 382 (1.1%) participants and in a mutually adjusted model including age and sex was significantly related to male sex, Muslim religion, having a mother or siblings who smoke, friends who smoke, low paternal education, low levels of family wealth and rebelliousness. The effects of exposure to smoking in films were then not statistically significant when added to this mutually adjusted model either as a binary exposure or in tertiles of intervals seen (Table 1).

A sensitivity analysis which removed variables that influenced the odds ratio for the effect of film exposure on incident smoking by less than 10% in the final mutually adjusted model (Model 2) in Table 1 generated a model that adjusted for gender, religion and friends smoking. The effects of film smoking, either as a binary or graded variable, remained similar in magnitude and non-significant when added to this more limited model.

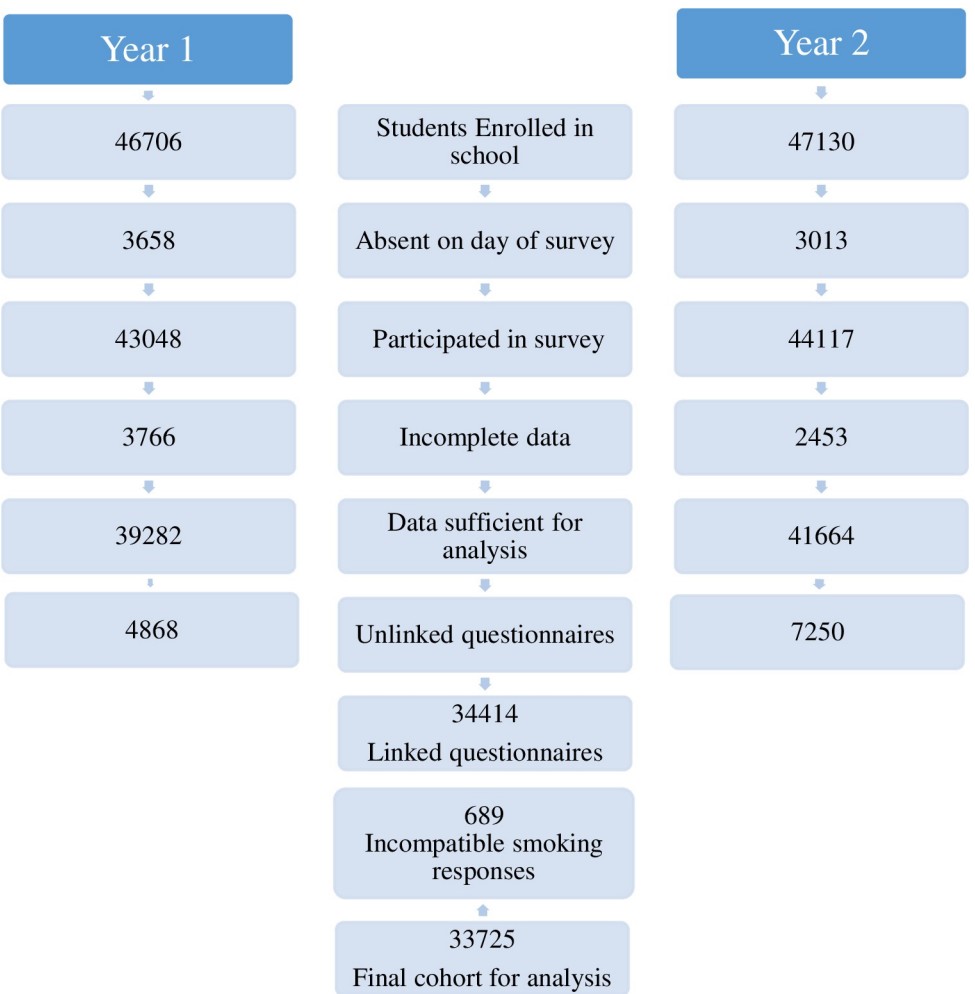

**Fig 1. Flow chart of participant numbers in the two surveys.**

Analysis of the effect of exposure to tobacco intervals in films according to the level of film compliance with COTPA Rules revealed no effect of exposure when added to the mutually adjusted model, in relation to whether films included AV disclaimers, health spots or the presence of any static health warning. However this analysis demonstrated that incident smoking was less common among those exposed to health warnings that complied fully with COTPA requirements to be legible, use approved text in the language of the film, and to be presented in black font on a white background (Table 2).

## Discussion

This first prospective study of incident smoking in young people from South India in a country which requires comprehensive anti-tobacco messaging before and during films containing smoking imagery finds that exposure to film tobacco imagery was not associated with increased smoking uptake. This confirms our earlier cross-sectional finding in this same cohort of children [12], and contrasts with a wealth of evidence from elsewhere in the world [3–7] and from a study carried out in India before the introduction of tobacco-free film rules in 2012 [22]. An exploratory analysis of compliance of the films seen by participants with the

**Table 1.** Demographic and environmental associations with incident smoking in the study population, including age and sex, showing univariate (Column 4) and statistically significant mutually adjusted odds ratios model before addition of effects of film tobacco exposure as either a binary or graded variable (Column 5 and Column 6).

| Characteristic | Number | Incident smokers (%) | Univariate OR (95% CI) | P value | Model1[+] - Adjusted OR[+] (95% CI) | P value | Model2[*] - Adjusted OR[*] (95% CI) | P value |
|---|---|---|---|---|---|---|---|---|
| **Age** | | | | 0.063[$] | | 0.394 | | 0.351 |
| 10 | 170(0.5%) | 2(1.2%) | 1 | | 1 | | 1 | |
| 11 | 5021(14.9%) | 53(1.1%) | 0.9(0.2,3.7) | | 1.2(0.3,4.9) | | 1.1(0.3,4.7) | |
| 12 | 11106(32.9%) | 141(1.3%) | 1.1(0.3,4.4) | | 1.4(0.3,5.9) | | 1.4(0.3,5.7) | |
| 13 | 11385(33.8%) | 117(1.0%) | 0.9(0.2,3.6) | | 1.2(0.3,5.0) | | 1.2(0.3,4.9) | |
| 14 | 5711(16.9%) | 60(1.1%) | 0.9(0.2,3.7) | | 1.2(0.3,5.0) | | 1.2(0.3,4.9) | |
| 15 | 332(1.0%) | 9(2.7%) | 2.3(0.5,11.0) | | 2.2(0.5,10.5) | | 2.2(0.5,10.5) | |
| **Gender** | | | | <0.001 | | <0.001 | | <0.001 |
| Female | 17134(50.8%) | 112(0.7%) | 1 | | 1 | | 1 | |
| Male | 16591(49.2%) | 270(1.6%) | 2.5(2.0,3.1) | | 2.3(1.8,2.8) | | 2.2(1.8,2.8) | |
| **School locality** | | | | 0.059 | | 0.015 | | |
| Rural | 27153(80.5%) | 293(1.1%) | 1 | | 1 | | | |
| Urban | 6572(19.5%) | 89(1.4%) | 1.3(1.0,1.6) | | 1.4(1.1,1.8) | | | |
| **School type** | | | | 0.493 | | 0.409 | | |
| Private | 12852(38.1%) | 135(1.1%) | 1 | | 1 | | | |
| Govt | 14438(42.8%) | 168(1.2%) | 1.1(0.9,1.4) | | 1.2(0.9,1.6) | | | |
| Aided | 6435(19.1%) | 79(1.2%) | 1.2(0.9,1.5) | | 1.2(0.9,1.7) | | | |
| **Religion** | | | | <0.001 | | <0.001 | | <0.001 |
| Hindu | 28444(84.3%) | 283(1.0%) | 1 | | 1 | | 1 | |
| Muslim | 3372(10.0%) | 70(2.1%) | 2.1(1.6,2.7) | | 2.2(1.6,2.9) | | 2.1(1.6,2.8) | |
| Other | 1909(5.7%) | 29(1.5%) | 1.5(1.0,2.3) | | 1.5(1.0,2.3) | | 1.5(1.0,2.3) | |
| **Home smoking allowed** | | | | <0.001 | | <0.001 | | <0.001 |
| No | 30724(91.1%) | 305(1.0%) | 1 | | 1 | | 1 | |
| Yes | 3001(8.9%) | 77(2.6%) | 2.6(2.0,3.4) | | 1.9(1.5,2.5) | | 1.9(1.5,2.5) | |
| **Family smoking** | | | | | | | | |
| Father No | 30354(90.0%) | 321(1.1%) | 1 | <0.001 | 1 | 0.149 | 1 | 0.131 |
| Yes | 3371(10.0%) | 61(1.8%) | 1.7(1.3,2.3) | | 1.3(0.9,1.7) | | 1.3(0.9,1.7) | |
| Mother No | 33491(99.3%) | 369(1.1%) | 1 | <0.001 | 1 | 0.004 | 1 | 0.003 |
| Yes | 234(0.7%) | 13(5.6%) | 5.3(3.0,9.3) | | 2.5(1.3,4.6) | | 2.5(1.4,4.7) | |
| Siblings No | 33235(98.5%) | 363(1.1%) | 1 | <0.001 | 1 | 0.028 | 1 | 0.024 |
| Yes | 490(1.5%) | 19(3.9%) | 3.7(2.3,5.8) | | 1.8(1.1,2.9) | | 1.8(1.1,2.9) | |
| others No | 28485(84.5%) | 318(1.1%) | 1 | 0.509 | 1 | 0.814 | | |
| Yes | 5240(15.5%) | 64(1.2%) | 1.1(0.8,1.4) | | 1.0(0.8,1.4) | | | |
| **Friends smoking** | | | | <0.001 | | <0.001 | | <0.001 |
| None | 30122(89.3%) | 291(1.0%) | 1 | | 1 | | 1 | |
| Anyone | 1151(3.4%) | 55(4.8%) | 5.1(3.8,6.9) | | 2.8(2.0,3.8) | | 2.8(2.0,3.8) | |
| Not sure | 2452(7.3%) | 36(1.5%) | 1.5(1.1,2.2) | | 1.2(0.8,1.7) | | 1.2(0.9,1.7) | |
| **Smoking seen in school** | | | | <0.001 | | 0.085 | | 0.063 |
| No | 24348(72.2%) | 242(1.0%) | 1 | | 1 | | 1 | |
| Yes | 9377(27.8%) | 140(1.5%) | 1.5(1.2,1.9) | | 1.2(1.0,1.5) | | 1.2(1.0,1.5) | |
| **Mothers education** | | | | 0.286[$] | | 0.983 | | |
| Non response | 480(1.4%) | 8(1.7%) | 1.9(0.8,4.6) | | 0.9(0.3,2.4) | | | |
| Illiterate | 1980(5.9%) | 30(1.5%) | 1.7(0.9,3.3) | | 0.9(0.4,2.1) | | | |
| Primary | 13942(41.3%) | 166(1.2%) | 1.3(0.7,2.4) | | 0.9(0.4,1.8) | | | |
| High school | 12512(37.1%) | 128(1.0%) | 1.1(0.6,2.1) | | 0.8(0.4,1.7) | | | |

*(Continued)*

**Table 1.** (Continued)

| Characteristic | Number | Incident smokers (%) | Univariate OR (95% CI) | P value | Model1[+] - Adjusted OR[+] (95% CI) | P value | Model2[*] - Adjusted OR[*] (95% CI) | P value |
|---|---|---|---|---|---|---|---|---|
| Graduate | 3476(10.3%) | 38(1.1%) | 1.2(0.6,2.3) | | 1.0(0.5,1.9) | | | |
| Postgraduate | 1335(4.0%) | 12(0.9%) | 1 | | 1 | | | |
| **Fathers education** | | | | 0.014[$] | | 0.113 | | 0.049 |
| Non response | 625(1.9%) | 14(2.2%) | 3.2(1.5,6.8) | | 3.6(1.5,8.8) | | 3.2(1.5,7.0) | |
| Illiterate | 1441(4.3%) | 24(1.7%) | 2.4(1.2,4.7) | | 2.1(0.9,4.9) | | 2.1(1.0,4.2) | |
| Primary | 13711(40.7%) | 162(1.2%) | 1.7(0.9,2.9) | | 1.8(0.9,3.6) | | 1.6(0.9,2.9) | |
| High school | 12572(37.3%) | 130(1.0%) | 1.5(0.8,2.6) | | 1.6(0.8,3.2) | | 1.5(0.8,2.6) | |
| Graduate | 3550(10.5%) | 39(1.1%) | 1.5(0.8,2.9) | | 1.6(0.8,3.2) | | 1.6(0.8,3.0) | |
| Postgraduate | 1826(5.4%) | 13(0.7%) | 1 | | 1 | | 1 | |
| **Wealth Quintile** | | | | <0.001[$] | | 0.020 | | 0.015 |
| Lower | 6746(20.0%) | 107(1.6%) | 1.2(0.9,1.6) | | 1.2(0.9,1.7) | | 1.2(0.9,1.7) | |
| Lower middle | 6913(20.5%) | 70(1.0%) | 0.8(0.6,1.1) | | 0.9(0.6,1.3) | | 0.9(0.6,1.3) | |
| Middle | 6593(19.5%) | 57(0.9%) | 0.7(0.5,0.9) | | 0.8(0.5,1.1) | | 0.8(0.5,1.1) | |
| Upper middle | 6891(20.4%) | 62(0.9%) | 0.7(0.5,1.0) | | 0.8(0.6,1.1) | | 0.8(0.5,1.1) | |
| Upper | 6582(19.5%) | 86(1.3%) | 1 | | 1 | | 1 | |
| **Rebelliousness** | | | | <0.001[$] | | <0.001 | | <0.001 |
| No | 21298(63.2%) | 198(0.9%) | 1 | | 1 | | 1 | |
| Mild | 9371(27.8%) | 86(0.9%) | 1.0(0.8,1.3) | | 1.0(0.8,1.3) | | 1.0(0.8,1.3) | |
| Moderate | 2728(8.1%) | 78(2.9%) | 3.1(2.4,4.1) | | 2.7(2.1,3.6) | | 2.7(2.1,3.6) | |
| Severe | 328(1.0%) | 20(6.1%) | 6.9(4.3,11.1) | | 5.2(3.2,8.5) | | 5.1(3.1,8.3) | |
| **High self-esteem** | | | | 0.100[$] | | 0.282 | | |
| Strongly Agree | 15675(46.5%) | 163(1.0%) | 1 | | 1 | | | |
| Agree | 7568(22.4%) | 89(1.2%) | 1.1(0.9,1.5) | | 1.1(0.8,1.4) | | | |
| Neither agree nor disagree | 5848(17.3%) | 63(1.1%) | 1.0(0.8,1.4) | | 1.0(0.7,1.3) | | | |
| Disagree | 2596(7.7%) | 43(1.7%) | 1.6(1.1,2.2) | | 1.5(1.0,2.1) | | | |
| Strongly Disagree | 2038(6.0%) | 24(1.2%) | 1.1(0.7,1.7) | | 0.9(0.6,1.5) | | | |
| **School Performance** | | | | 0.001[$] | | 0.101 | | 0.102 |
| Excellent | 13119(38.9%) | 154(1.2%) | 1 | | 1 | | 1 | |
| Good | 15567(46.2%) | 146(0.9%) | 0.8(0.6,1.0) | | 0.8(0.6,1.0) | | 0.8(0.6,1.0) | |
| Average | 4340(12.9%) | 68(1.6%) | 1.3(1.0,1.8) | | 1.0(0.7,1.4) | | 1.0(0.8,1.4) | |
| Below Average | 699(2.1%) | 14(2.0%) | 1.7(1.0,3.0) | | 1.1(0.6,1.9) | | 1.1(0.6,1.9) | |
| **Film tobacco exposure (binary)** | | | | 0.256 | | 0.493 | | 0.437[@] |
| No | 506(1.5%) | 3(0.6%) | 1 | | 1 | | 1 | |
| Yes | 33219(98.5%) | 379(1.1%) | 1.9(0.6,6.0) | | 1.5(0.5,4.7) | | 1.6(0.5,4.9) | |
| **Film tobacco exposure (tertiles)** | | | | 0.006[$] | | 0.273 | | 0.228[@] |
| 0 | 506(1.5%) | 3 (0.6%) | 1 | | 1 | | 1 | |
| 1–49 | 10670(31.6%) | 121 (1.1%) | 1.9(0.6,6.1) | | 1.6(0.5,5.0) | | 1.6(0.5,5.2) | |
| 50–84 | 11580(34.3%) | 106 (0.9%) | 1.5(0.5,4.9) | | 1.3(0.4,4.1) | | 1.4(0.4,4.3) | |
| >84 | 10969(32.5%) | 152 (1.4%) | 2.4(0.7,7.4) | | 1.6(0.5,5.1) | | 1.7(0.5,5.5) | |

[+] Model 1 includes all variables with mutual adjustment, irrespective of significance.

[*] Model 2 includes age, sex and variables then independently associated with smoking uptake before addition of film tobacco exposure.

[@] Smoking in films as binary, and as trend through quintiles were not included in the same model.

[$] P value for trend.

**Table 2. Effect of tobacco free film rules on incident smoking, added to the mutually adjusted model in Table 1 in place of film tobacco variables, graded according to whether the film contained audio-visual disclaimers, health spots and static warning messages either in any form or in form compliant with legal requirements.**

| Smoke free film rule requirement | Number (%) | Incident smokers (%) | Univariate OR (95% CI) | p value | Model1*-Adjusted *OR (95% CI) | p value | Model 2*-Adjusted *OR (95% CI) | p value |
|---|---|---|---|---|---|---|---|---|
| **AV disclaimer at the start of the film** | | | | 0.634 | | 0.756 | | 0.859 |
| Yes | 27032 (80.2%) | 312(1.2%) | 1.1(0.8,1.4) | | 1.0(0.7,1.3) | | 1.0(0.7,1.3) | |
| No | 6187 (18.3%) | 67(1.1%) | 1 | | 1 | | 1 | |
| **Health Spot at the start of the film** | | | | 0.114 | | 0.186 | | 0.153 |
| Yes | 31064 (92.1%) | 362(1.2%) | 1.5(0.9,2.4) | | 1.4(0.9,2.3) | | 1.4(0.9,2.4) | |
| No | 2155(6.4%) | 17(0.8%) | 1 | | 1 | | 1 | |
| **Health Spot in the middle of the film** | | | | 0.199 | | 0.234 | | 0.170 |
| Yes | 30421 (90.2%) | 354(1.2%) | 1.3(0.9,2.0) | | 1.3(0.9,2.0) | | 1.3(0.9,2.0) | |
| No | 2798(8.3%) | 25(0.9%) | 1 | | 1 | | 1 | |
| **Static warning messages** | | | | 0.608 | | 0.612 | | 0.580 |
| Yes | 33023 (97.9%) | 376(1.1%) | 0.7(0.2,2.3) | | 0.7(0.2,2.4) | | 0.7(0.2,2.3) | |
| No | 196(0.6%) | 3(1.5%) | 1 | | 1 | | 1 | |
| **COTPA-compliant static warning messages** | | | | 0.012 | | 0.014 | | 0.031 |
| Yes | 26437 (78.4%) | 282(1.1%) | 0.7(0.6,0.9) | | 0.7(0.6,0.9) | | 0.8(0.6,1.0)** | |
| No | 6782 (20.1%) | 97(1.4%) | 1 | | 1 | | 1 | |

Model1*- Each of the variables was independently adjusted for age, gender, School locality, School type, Religion, Family smoking, Friends smoking, Smoking seen in school, Mother Education, Father Education, wealth quintile, rebelliousness, High self-esteem and School performance.

Model2*—Each of the variables was independently adjusted for age, gender, religion, father's education, family members smoking, friends smoking, wealth quintile and rebelliousness.

** figures in table rounded to one decimal place; unrounded figures = 0.76 (0.59 to 0.98).

requirements of the 2012 Rules [11] found evidence that the presence of on-screen health warnings that used approved text, were clearly legible, in black print on a white background, in the language used in the film were associated with less incident smoking, but did not confirm the protective effect of audio-visual disclaimers found in our baseline cross-sectional analysis of this population [12].

This study was carried out to try to quantify the effect of exposure to tobacco imagery in film on incident smoking in Southern India, and our null finding was therefore unexpected. However, our study used a large sample of children in grades 7–9 from Udupi district in Karnataka State, and measures of film imagery exposure derived from the most popular films in Karnataka in the two years preceding the study. The proportion of eligible children participating in our follow up of the cohort was extremely high, with usable linked data from the two study years collected from over 70% of all children eligible for inclusion at the outset of the study, and 86% of those who provided usable data in the first study year [12]. Although the number of incident smokers in our study was small, our study was powered to detect an effect of film exposure of similar magnitude to that seen in other countries. Although we cannot rule

out a false negative finding therefore, our analysis of factors associated with smoking uptake in the present study, aside from those relating to film smoking, were similar to those reported to be risk factors in studies from elsewhere in the world [5]. It is therefore highly likely that our findings are representative and, to the extent that children in Karnataka are representative of children living elsewhere in India, generalisable.

We are not aware of any previous prospective studies of the effect of films containing anti-smoking messages on smoking uptake in young people. Earlier work has demonstrated that in audiences including children, the showing of anti-smoking messages before films containing smoking generates higher awareness and lower levels of approval of smoking in the film, and more negative attitudes towards smoking in general compared with those not exposed [9], and changes the perceived status of smoking from that of a forbidden fruit to something that is tainted [10]. However, generating these perceptions at the time the film is shown does not necessarily translate into a change in future smoking behaviour. Our study, while far from conclusive, suggests however that this translation can indeed occur. While we acknowledge that our finding on on-screen health warnings applied to only one of the several tobacco-free film measures analysed, it is also plausible that antismoking messages appearing on screen during smoking content would have this effect.

Tobacco uptake among adolescents arises from synergistic interplay of various socio-cultural and environmental aspects. It has been demonstrated in our study that male gender; those with declining family wealth; sibling and friends smoking; and low paternal education have all played a causal role in smoking uptake, and these findings are entirely consistent with the extensive existing international and Indian evidence base [4,5,16,23]. The present study has shown that the impact of film smoking imagery is likely to be attenuated by incorporating health warning messages in the film as explained above. However, it is important to address other social determinants of smoking uptake by developing a comprehensive policy with social norms and support in favor of tobacco control [24]. This would further lead to sustainable behaviour change and go a long way in reducing tobacco use in the society.

Smoking tobacco is highly addictive, lethal and entirely avoidable. Therefore, whilst our findings do not provide conclusive proof that anti-tobacco messaging is effective in preventing exposure to film tobacco imagery from causing smoking uptake, or exclude the possibility that our findings arise from unrecognized confounding by other tobacco policy measures, the fact that the messaging required under Indian Rules is simple and inexpensive to implement, and can be applied universally to all films screened, broadcast or streamed and irrespective of the age-rating of the film, makes a compelling argument to apply the Indian approach more widely. Despite a trend towards reduced inclusion of tobacco imagery over time in films popular elsewhere in the world, and undertakings by some major US studios to remove smoking from films aimed at children, smoking remains common in films classified as suitable for viewing by children and young people [25–27]. Global adoption of Indian tobacco-free film and TV rules [11] offers a potential means of smoking prevention that is inexpensive and universally applicable. Therefore, while further study of the optimal design and content of such measures would be welcome, we advocate their immediate and widespread adoption.

## Supporting information

**S1 Annexure. Key study variables.**
(DOCX)

**S1 File. Questionnaires.**
(ZIP)

## Acknowledgments

The authors thank the Deputy Director of Public Instructions of Udupi district for the permission, all the students who participated in the survey and the teachers for helping in the coordination. They also acknowledge the contributions of the research assistants and data collectors for their meticulous data collection.

## Author Contributions

**Conceptualization:** Muralidhar M. Kulkarni, Veena G. Kamath, Monika Arora, John Britton.

**Data curation:** Muralidhar M. Kulkarni, Asha Kamath, Sarah Lewis, Manpreet Bains, Monika Arora, John Britton.

**Formal analysis:** Muralidhar M. Kulkarni, Asha Kamath, Veena G. Kamath, Monika Arora, John Britton.

**Funding acquisition:** Muralidhar M. Kulkarni, John Britton.

**Investigation:** Muralidhar M. Kulkarni, Asha Kamath, Veena G. Kamath, Ilze Bogdanovica, Monika Arora, John Britton.

**Methodology:** Muralidhar M. Kulkarni, Asha Kamath, Veena G. Kamath, Ilze Bogdanovica, Andrew Fogarty, Monika Arora, Gaurang P. Nazar, John Britton.

**Project administration:** Muralidhar M. Kulkarni, Veena G. Kamath, Monika Arora, John Britton.

**Resources:** Muralidhar M. Kulkarni, Andrew Fogarty, Gaurang P. Nazar, Kirthinath Ballal, John Britton.

**Software:** Asha Kamath, Jo Cranwell, Gaurang P. Nazar, Kirthinath Ballal, Ashwath K. Naik.

**Supervision:** Muralidhar M. Kulkarni, Kirthinath Ballal, Rohith Bhagawath.

**Validation:** Veena G. Kamath, Sarah Lewis, Manpreet Bains, Jo Cranwell, Gaurang P. Nazar, Kirthinath Ballal.

**Visualization:** Muralidhar M. Kulkarni, Monika Arora, Gaurang P. Nazar, Rohith Bhagawath.

**Writing – original draft:** Muralidhar M. Kulkarni, Asha Kamath, Veena G. Kamath, John Britton.

**Writing – review & editing:** Asha Kamath, Sarah Lewis, Ilze Bogdanovica, Manpreet Bains, Jo Cranwell, Andrew Fogarty, Monika Arora, Gaurang P. Nazar, Kirthinath Ballal, Rohith Bhagawath.

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
