## [Decision Letter · Decision Letter 0]

27 Jan 2021

PONE-D-20-33310

Prospective cohort study of exposure to tobacco imagery in popular films and smoking uptake among children in southern India

PLOS ONE

Dear Dr. Kamath,

Thank you for submitting your manuscript to PLOS ONE. After careful consideration, we feel that it has merit but does not fully meet PLOS ONE’s publication criteria as it currently stands. Therefore, we invite you to submit a revised version of the manuscript that addresses the points raised during the review process.

Two reviewers with significant expertise in public health and in tobacco control studies have provided their comments and assessment of your manuscript. Based on my own reading, I concur with the reviewers' recommendation to reconsider the way the data is presented in Tables, and the way it is interpreted, especially with regards to the null effects. I hope that you will find the reviewers' comments useful and constructive.

We look forward to receiving your revised manuscript.

Kind regards,

Lambros Lazuras

Academic Editor

PLOS ONE

Journal Requirements:

2. Please include additional information regarding the survey or questionnaire used in the study and ensure that you have provided sufficient details that others could replicate the analyses.

For instance, if you developed a questionnaire as part of this study and it is not under a copyright more restrictive than CC-BY, please include a copy, in both the original language and English, as Supporting Information.

4. Thank you for stating the following financial disclosure: 'The funders had no role in study design, data collection and analysis, decision to publish, or preparation of the manuscript.'

Reviewers' comments:

Reviewer's Responses to Questions

**Comments to the Author**

1. Is the manuscript technically sound, and do the data support the conclusions?

Reviewer #1: Yes

Reviewer #2: No

2. Has the statistical analysis been performed appropriately and rigorously? 

Reviewer #1: Yes

Reviewer #2: No

3. Have the authors made all data underlying the findings in their manuscript fully available?

Reviewer #1: Yes

Reviewer #2: Yes

4. Is the manuscript presented in an intelligible fashion and written in standard English?

Reviewer #1: Yes

Reviewer #2: No

5. Review Comments to the Author

Reviewer #1: It was a pleasure to read this well written paper. The researchers have a large data set that assesses the risk of taking up smoking over two years by analysing a variety of home and exposure to smoking in films. I have a few comments that will improve the manuscript.

1. as PLOS One has no word count more description of the explanatory variables is needed rather than referring to a previous publication. Research papers should, as much as possible, stand alone. I realise that the less well explained variables are the control variables but the same level of detail is needed to assist in interpreting the tables.

2. I’d be happier seeing all the variables in the final model to see how adding the film exposure variables affects the control variables. This is a matter of convention and in environmental health the convention is to report them all. This is, essentially a report about the impact of environmental factors.

3. Be more honest in the discussion as only 1 of 5 measures of film exposure was statistically significant. Given the magnitude of the home effects it is surprising that anything from the wider environment is statistically significant. Just don’t over-egg the finding.

4. The discussion should bring in more literature about the how home and wider social factors intersect. There will be research about this in sociological literature.

5. There does not seem to be a section in the discussion on limitations of the study. One of these is the relatively small number of students who took up smoking. Was there a sample size calculation in the original research plan?

I hope you are able to find funding to continue following up this cohort.

Reviewer #2: Unfortunately, I still believe that the presentation of the study findings remain incomprehensible for the general reader. Table 1 should show percentages along with frequencies. Table 3 remain incomprehensible for the general reader.

6. PLOS authors have the option to publish the peer review history of their article (what does this mean?). If published, this will include your full peer review and any attached files.

Reviewer #1: **Yes: **Professor Shona J. Kelly

Reviewer #2: **Yes: **Elpidoforos S. Soteriades

---

## [Author Response · Author response to Decision Letter 0]

12 Mar 2021

Journal Requirements:

Authors Reply: The authors have reviewed the manuscript thoroughly to ensure it meets the PLOS ONE’s style requirement.

2. Please include additional information regarding the survey or questionnaire used in the study and ensure that you have provided sufficient details that others could replicate the analyses.

For instance, if you developed a questionnaire as part of this study and it is not under a copyright more restrictive than CC-BY, please include a copy, in both the original language and English, as Supporting Information.

Authors Reply: The questionnaire in the original language and in English has been included as supporting information.

We will update your Data Availability statement on your behalf to reflect the information you provide. The data may

Authors Reply: The data will be made available on reasonable request as per the guidelines of institutional ethics committee by contacting the below mentioned email address by researchers who meet the criteria by access to confidential data. (Email: mrcuk.antitobacco@manipal.edu)

4. Thank you for stating the following financial disclosure: 'The funders had no role in study design, data collection and analysis, decision to publish, or preparation of the manuscript.'

Please clarify the sources of funding (financial or material support) for your study. List the grants or organizations that supported your study, including funding received from your institution.

State what role the funders took in the study. If the funders had no role in your study, please state: “The funders had no role in study design, data collection and analysis, decision to publish, or preparation of the manuscript.”

If any authors received a salary from any of your funders, please state which authors and which funders.

If you did not receive any funding for this study, please state: “The authors received no specific funding for this work.”

Authors Reply: The authors are fulltime employees of their respective institutions and draw incentives to compensate for the time through the grant. Dr. Rohit Bhagawath is the Social Scientist in the project and gets his salary from the project and not from the host institution.

Authors Reply: The grant numbers received for our study in the ‘Funding Information’ section has been verified.

Reviewers' comments:

Reviewer's Responses to Questions

Comments to the Author

1. Is the manuscript technically sound, and do the data support the conclusions?

Reviewer #1: Yes

Reviewer #2: No

Authors Reply: In this revised submission we have attempted to address Reviewer #2’s concerns in these areas. 

2. Has the statistical analysis been performed appropriately and rigorously?

Reviewer #1: Yes

Reviewer #2: No

Authors Reply: As above, we have endeavoured to resolve all of Reviewer #2’s concerns over our analysis. 

3. Have the authors made all data underlying the findings in their manuscript fully available?

Reviewer #1: Yes

Reviewer #2: Yes

Authors Reply: None necessary. 

4. Is the manuscript presented in an intelligible fashion and written in standard English?

Reviewer #1: Yes

Reviewer #2: No

Authors Reply: Again, we have endeavoured to resolve all of Reviewer #2’s concerns over intelligibility and use of English. 

5. Review Comments to the Author

Reviewer #1: It was a pleasure to read this well written paper. The researchers have a large data set that assesses the risk of taking up smoking over two years by analysing a variety of home and exposure to smoking in films. I have a few comments that will improve the manuscript.

1. as PLOS One has no word count more description of the explanatory variables is needed rather than referring to a previous publication. Research papers should, as much as possible, stand alone. I realise that the less well explained variables are the control variables but the same level of detail is needed to assist in interpreting the tables.

Authors reply: Thank you for the suggestion. We have now provided a more complete description of all the variables, but to maintain the readability of the paper have provided these in an Annexe document. 

2. I’d be happier seeing all the variables in the final model to see how adding the film exposure variables affects the control variables. This is a matter of convention and in environmental health the convention is to report them all. This is, essentially a report about the impact of environmental factors. 

Authors reply: We have now added additional model to both the tables that includes all the variables irrespective of their significance along with film tobacco imagery. 

3. Be more honest in the discussion as only 1 of 5 measures of film exposure was statistically significant. Given the magnitude of the home effects it is surprising that anything from the wider environment is statistically significant. Just don’t over-egg the finding.

Authors reply: We have again analysed the data creating another model as described above and the results have been provided in both the tables. 

4. The discussion should bring in more literature about the how home and wider social factors intersect. There will be research about this in sociological literature.

Authors reply: Thank you for the suggestion. We have now added a paragraph in the discussion to address this concern of the reviewer. 

5. There does not seem to be a section in the discussion on limitations of the study. One of these is the relatively small number of students who took up smoking. Was there a sample size calculation in the original research plan?

I hope you are able to find funding to continue following up this cohort.

Authors reply: 

We have added the below calculation in methods section as we had proposed in the study protocol in the methods section. 

Reviewer #2: 

6. Unfortunately, I still believe that the presentation of the study findings remains incomprehensible for the general reader. Table 1 should show percentages along with frequencies. Table 3 remain incomprehensible for the general reader.

Authors reply: Thank you for the suggestion. We have simplified the language wherever possible. We have included percentages along with frequencies in Table 1. We have provided explanation for all the variables in the Annexure attached as supplementary file. 

6. PLOS authors have the option to publish the peer review history of their article (what does this mean?). If published, this will include your full peer review and any attached files.

Do you want your identity to be public for this peer review? For information about this choice, including consent withdrawal, please see our Privacy Policy.

Reviewer #1: Yes: Professor Shona J. Kelly

Reviewer #2: Yes: Elpidoforos S. Soteriades

---

## [Decision Letter · Decision Letter 1]

9 Jun 2021

Prospective cohort study of exposure to tobacco imagery in popular films and smoking uptake among children in southern India

PONE-D-20-33310R1

Dear Dr. Kamath,

We’re pleased to inform you that your manuscript has been judged scientifically suitable for publication and will be formally accepted for publication once it meets all outstanding technical requirements.

Kind regards,

Jagdish Khubchandani

Academic Editor

PLOS ONE

Additional Editor Comments (optional):

Reviewers' comments:

Reviewer's Responses to Questions

**Comments to the Author**

1. If the authors have adequately addressed your comments raised in a previous round of review and you feel that this manuscript is now acceptable for publication, you may indicate that here to bypass the “Comments to the Author” section, enter your conflict of interest statement in the “Confidential to Editor” section, and submit your "Accept" recommendation.

Reviewer #1: All comments have been addressed

2. Is the manuscript technically sound, and do the data support the conclusions?

Reviewer #1: Yes

3. Has the statistical analysis been performed appropriately and rigorously? 

Reviewer #1: Yes

4. Have the authors made all data underlying the findings in their manuscript fully available?

Reviewer #1: Yes

5. Is the manuscript presented in an intelligible fashion and written in standard English?

Reviewer #1: Yes

6. Review Comments to the Author

Reviewer #1: I still think you are over 'egging' the barely positive findings but the confidence intervals show how borderline the results were.

7. PLOS authors have the option to publish the peer review history of their article (what does this mean?). If published, this will include your full peer review and any attached files.

Reviewer #1: **Yes: **Shona Kelly

---

## [Editor Report · Acceptance letter]

27 Jul 2021

PONE-D-20-33310R1 

Prospective cohort study of exposure to tobacco imagery in popular films and smoking uptake among children in southern India 

Dear Dr. Kamath:

I'm pleased to inform you that your manuscript has been deemed suitable for publication in PLOS ONE. Congratulations! Your manuscript is now with our production department. 

Kind regards, 

on behalf of

Dr. Jagdish Khubchandani 

Academic Editor

PLOS ONE